# Pathophysiological Involvement of Mast Cells and the Lipid Mediators in Pulmonary Vascular Remodeling

**DOI:** 10.3390/ijms24076619

**Published:** 2023-04-01

**Authors:** Hidenori Moriyama, Jin Endo

**Affiliations:** 1Department of Cardiology, Keio University School of Medicine, 35 Shinanomachi, Shinjuku 160-8582, Tokyo, Japan; 2Department of Cardiology, Tokyo Dental College Ichikawa General Hospital, 5-11-13 Sugano, Ichikawa 272-8513, Chiba, Japan

**Keywords:** pulmonary hypertension, lipid mediator, mast cell, remodeling, omega-3, n-3 epoxides

## Abstract

Mast cells are responsible for IgE-dependent allergic responses, but they also produce various bioactive mediators and contribute to the pathogenesis of various cardiovascular diseases, including pulmonary hypertension (PH). The importance of lipid mediators in the pathogenesis of PH has become evident in recent years, as exemplified by prostaglandin I2, the most central therapeutic target in pulmonary arterial hypertension. New bioactive lipids other than eicosanoids have also been identified that are associated with the pathogenesis of PH. However, it remains largely unknown how mast cell-derived lipid mediators are involved in pulmonary vascular remodeling. Recently, it has been demonstrated that mast cells produce epoxidized n-3 fatty acid (n-3 epoxides) in a degranulation-independent manner, and that n-3 epoxides produced by mast cells regulate the abnormal activation of pulmonary fibroblasts and suppress the progression of pulmonary vascular remodeling. This review summarizes the role of mast cells and bioactive lipids in the pathogenesis of PH. In addition, we introduce the pathophysiological role and therapeutic potential of n-3 epoxides, a mast cell-derived novel lipid mediator, in the pulmonary vascular remodeling in PH. Further knowledge of mast cells and lipid mediators is expected to lead to the development of innovative therapies targeting pulmonary vascular remodeling.

## 1. Introduction

Mast cells, the main inflammatory cells in type I allergy and anaphylaxis, are activated when IgE that is bound to the high-affinity IgE receptor FcεRI on the surface is cross-linked with a specific antigen, releasing histamine and proteases in the process of degranulation. Additionally, lipid mediators such as leukotrienes and prostaglandins are subsequentially produced, causing an immediate-phase allergic reaction. In recent years, it has become clear that mast cells are influenced by the microenvironment of tissues and contribute to tissue-specific immune responses and the maintenance of local homeostasis. Furthermore, they are involved in pathologies other than allergy, including cardiovascular diseases [1,2,3]. Although there is still little evidence, bioactive lipids produced by mast cells have been suggested to be important molecules that affect cardiovascular disease.

Pulmonary arterial hypertension (PAH) is an intractable disease with a poor prognosis in which pulmonary vascular resistance increases because of peripheral pulmonary artery lumen narrowing, leading to right heart failure and death. The histopathological features of PAH are characterized by “pulmonary vascular remodeling”, such as the infiltration of inflammatory cells and the uncontrolled proliferation of endothelial cells, smooth muscle cells, and fibroblasts that constitute pulmonary vessels [4,5,6,7,8]. TGF/BMP signaling imbalance is well-known to be involved in the pulmonary vascular abnormalities seen in PAH, and BMPR2 gene mutations are the most common genetic cause of PAH. In addition, various factors such as epigenetic changes, immune changes, increased inflammation, and changes in mitochondrial metabolic function have been reported as pathogenic mechanisms of PAH [9]. Advances in the treatment using pulmonary vasodilators have improved the prognosis of this disease, but there is still no therapeutic method that suppresses the pulmonary vascular remodeling.

Various inflammatory cells are implicated in the progression of pulmonary vascular remodeling [10]. Mast cells are known to produce various humoral factors to induce tissue remodeling and are presumed to contribute to the progression of pulmonary hypertension (PH) pathology because they accumulate around pulmonary blood vessels in patients with PAH. However, considering the total number of inflammatory cells present in PH lungs, mast cells account for only a small proportion, and PH is not included in the category of allergic diseases, so the relationship between mast cells and PH has been poorly investigated and remains unclear.

In this manuscript, we broadly collected and systematically reviewed the current knowledge on the role of mast cells and bioactive lipids and their molecular mechanisms in the pathogenesis of PH with an aim to explore the direction for future research and therapeutic development in this field. First, we describe the contribution of mast cells in pulmonary vascular remodeling and then review the relationship between PH and bioactive lipids. In addition, we introduce the pathophysiological role and therapeutic potential of epoxidized n-3 fatty acid (n-3 epoxides), a mast cell-derived novel lipid mediator, in pulmonary vascular remodeling in PH.

## 2. Role of Mast Cells in Pulmonary Vascular Remodeling

Recently, mast cells have been implicated in the defense against allergic diseases and infection of parasites, but also in the development of cardiovascular diseases. Like other inflammatory cells, mast cells play multiple roles in the development of cardiometabolic diseases such as obesity, diabetes, and atherosclerosis and are associated with the complications of those diseases [11]. Mast cells are increased in tissues such those of the coronary arteries, heart, aorta, and adipose tissue in patients with cardiovascular diseases, and when activated secrete various humoral factors affecting disease pathogenesis [1,12]. Moreover, mast cells were associated with various biological processes by interacting with nearby vascular cells such as endothelial cells, fibroblasts, and smooth muscle cells, and with perivascular inflammatory cells, such as macrophages and lymphocytes.

The evidence that mast cells play an important role in the pathogenesis of PH is gradually increasing (Figure 1). The accumulation of mast cells around pulmonary vessels has been observed histopathologically in patients with PAH and in a rodent model of PH [13,14,15]. In addition, using comprehensive gene expression data or by flow cytometric analysis of biological samples from PAH patients and the latest bioinformatic techniques, it has been revealed that the development of PH is associated with the activation of mast cells [16,17,18]. A study from the 1970s reported a correlation between the number of mast cells and the severity of PH in various animal models of chronic hypoxic PH [19]. In the aortic banding model of Ws/Ws mast cell-deficient rats with a small deletion in the tyrosine kinase domain of the c-kit gene, PH, and pulmonary vascular remodeling in left-sided heart failure were attenuated compared to control rats, suggesting that the presence of mast cells may exacerbate the pathology of PH [20]. Further, several studies have shown that granule contents released by mast cells are deeply involved in development of PH [21,22,23]. Mast cell stabilizers that inhibit degranulation, such as cromolyn and disodium cromoglycate (DSCG), improve pulmonary vascular remodeling in various PH rat models [13,24,25]. In a small clinical trial, nine patients with PH were treated with mast cell inhibitors cromolyn and fexofenadine over 12 weeks resulting in decreased indicators of mast cell activation such as serum total tryptase and urine leukotriene E4. Furthermore, plasma vascular endothelial growth factor (VEGF) and circulating proangiogenic CD34 + CD133 + progenitor cell levels were reduced and exhaled nitric oxide was increased after the treatment, suggesting that mast cell products may contribute to the pathogenic proangiogenic environment in the pulmonary vascular remodeling of PH [26].

Mediators produced by mast cells such as various proteases and cytokines are involved in PH. For example, chymase, one of the major proteases secreted by mast cells, can lead to the local production of angiotensin II and the activation of endothelins and matrix metalloproteinases (MMPs). Chymase inhibition attenuated PH and pulmonary vascular remodeling and its effects were associated with a strong tendency toward a reduction in mast cell activity [24,27]. Tryptase, another protease contained in the granule contents of mast cells, induces the proliferation and migration of pulmonary artery smooth muscle cells as well as the synthesis of fibronectin and MMP-2 in a protease-activated receptor (PAR)-2- and extracellular signal-regulated kinase (ERK) 1/2-dependent manner [28]. IL-6, the major cytokine released by mast cells, has been reported to promote the recruitment of B cells and the subsequent production of antibodies, contributing to the development of PH. In experimental models of PH, mast cell stabilization as well as B cell depletion by treatment with an anti-IL-6 neutralizing antibody or anti-CD20 antibody reduced pulmonary vascular remodeling and improved PH [29].

Furthermore, tyrosine kinase inhibitors, such as imatinib, have also been noted to modify the pathogenesis of PH via mast cells. Administration of imatinib, a Bcr-Abl tyrosine kinase inhibitor, or sorafenib, a multiple tyrosine kinase inhibitor that targets both Raf and VEGF and platelet-derived growth factor (PDGF) receptor tyrosine kinase signaling, improved experimental PH in animal models [30,31,32]. Therefore, these inhibitors have attracted attention as candidates for next-generation drugs for PH treatment. The inhibitory effect of imatinib on pulmonary vascular remodeling is thought to be mainly because of its antagonistic action that inhibits the phosphorylation of PDGF receptors, thereby suppressing the proliferation of vascular smooth muscle cells. It is also pointed out that these drugs affected mast cells. As the growth of mast cells and the production and secretion of their granules are controlled by signaling mediated by c-kit tyrosine kinase receptor (CD117) [33], the tyrosine kinase inhibitors are thought to block the signal from CD117, inhibiting the number and function of mast cells to suppress the development of PH. In fact, in the Phase III trial for idiopathic PAH patients, Imatinib improved the results of the 6 min walk test and pulmonary artery pressure [34]. However, many patients experienced serious side effects and were unable to continue the treatment so, it has not yet been applied clinically [35].

Many studies have already confirmed that mast cells and their mediators are involved in the exacerbation and progression of PH. However, the underlying mechanisms are still elusive, and it may be premature to conclude that mast cells have an overall negative effect on PH. In fact, several studies using mast cell-deficient mice and mast cell stabilizers have shown different results suggesting that mast cells can be beneficial in the pathology of PH, depending on the experimental conditions. For example, in the chronic hypoxia-exposed PH model of W/Wv mice, which are congenitally deficient in mast cells, pulmonary vascular remodeling and increased right ventricular weight were attenuated compared with hypoxic PH wild-type mice [36]. Additionally, administration of DSCG to rats exposed to hypoxia failed to suppress right ventricular hypertrophy [37]. Therefore, simply suppressing the number or activation of mast cells may not necessarily have a beneficial effect on PH.

Degranulation is thought to be a major mechanism by which mast cells influence PH pathology, but it is unclear whether degranulation in PH has the same mechanism of action or the same granule contents as in an immediate-type allergy. The degranulation of mast cells induced by hypoxia, one of the triggers of PH, is known to be mediated by transient receptor potential ankyrin 1 (TRPA1) [3], and its morphological changes have also been reported to differ in the degranulation mediated by IgE [38]. In addition, not only IgE but also various agents such as IgG, complement factors, Toll-like receptor peptides, and substance P affect the activation of mast cells [39,40,41,42,43], but it is not clear what changes occur in the amount or composition of the granule contents released from mast cells. Furthermore, it is necessary to investigate the effects of functions of mast cells other than the degranulation of mast cells on PH. In recent years, it has been reported that mast cells have not only proinflammatory properties such as allergic inflammation, but also anti-inflammatory and immunosuppressive functions [44,45,46]. When mast cells exert negative immunomodulatory functions in vivo, mast cell-derived products, particularly IL-10, limit specific acquired immune responses and the innate immune response to chronic irradiation with UVB light, independently of degranulation.

## 3. Bioactive Lipid Mediators Involved in Pulmonary Vascular Remodeling

Bioactive lipids are enzymatically produced not only in inflammatory cells but also in various lung tissue cells and exert their unique functions through binding to specific receptors on extracellular targets. Each bioactive lipid has a diverse function and plays a central role in tissue remodeling by regulating processes including inflammation, thrombus formation, angiogenesis, and fibrosis. Lysophospholipids and free fatty acids are produced from phospholipids in cell membranes by hydrolysis of phospholipase A2, and they or their metabolites function as active mediators [47]. Fatty acids, especially polyunsaturated fatty acids (PUFA) represented by arachidonic acid (AA), are sequentially metabolized by oxidases, such as cyclooxygenase (COX) and lipoxygenase (LOX), and prostaglandin synthase to acquire strong activity and specific function. In this chapter, we review the major bioactive lipids associated with PH that have been reported to date (Figure 1).

### 3.1. Lysophospholipid (Lyso PL)

#### 3.1.1. Sphingosine-1 Phosphate (S1P)

Sphingosine-1 phosphate (S1P), a lysophospholipid derived from sphingolipids, is thought to contribute to the pathogenesis of PH because it is a functional lipid involved in cell proliferation, migration, and angiogenesis [48,49]. Sphingosine kinase 1 (Sphk1), the major enzyme responsible for the production of S1P, was upregulated in the lung tissue and vascular smooth muscle cells of patients with PAH and in the lung vessels of hypoxia-exposed PH mice. In addition, gene deletion of Sphk1 or administration of Sphk1 inhibitors attenuated the PH phenotype [50]. Sphk1/S1P played a pivotal role in the abnormal proliferation of pulmonary artery smooth muscle cells through Notch signaling when stimulated with TGF-β1 or through miR-21/BMPRII/Id1 pathway when stimulated with PDGF [51,52].

#### 3.1.2. Lysophosphatidic Acid (LPA)

Lysophosphatidic acid (LPA) is a lysophospholipid produced by plasma lysophospholipase D (autotaxin). LPA induces proinflammatory stimuli for a variety of vascular cells, the activation of human platelets, the proliferation and migration of smooth muscle cells, and the dysfunction of endothelial cells via LPA-selective G-protein coupled receptors, suggesting the potential for involvement in a variety of vascular diseases [53]. In experiments using heterozygous autotaxin-deficient mice or LPA1,2-receptors double KO mice, contrary to expectations, the phenotype of hypoxia-induced PH was rather severe, indicating that LPA may negatively regulate pulmonary vascular pressure and vascular remodeling through LPA1 and LPA2 receptors [54]. The role of LPA in PH is still largely unclear.

### 3.2. Arachidonic Acid (AA) Metabolites

#### 3.2.1. Prostaglandin I2 (PGI2)

PGI2 is the lipid mediator that plays the most central role in the pathogenesis of PH and has been established as a therapeutic target. PGI2 is the eicosanoid enzymatically produced from AA by COX-2 and prostaglandin I synthase in pulmonary endothelial cells. PGI2 inhibits platelet aggregation and relaxes the tone of vascular smooth muscle by stimulating cAMP production via the PGI2 receptor IP expressed in platelets, endothelial cells, and vascular smooth muscle cells [55]. In contrast, thromboxane A2 (TXA2), one of the proinflammatory eicosanoids, functions to promote strong pulmonary vasoconstriction and activating platelet aggregation. In PAH, the balance of vasoactive substances is disrupted because of vascular endothelial dysfunction, resulting in an increased production of the vasoconstrictor TXA2 and decreased production of the vasodilator PGI2, inducing aberrant vasocontraction of the pulmonary artery and abnormal proliferation of pulmonary smooth muscle cells [56]. In the 1980s, administration of PGI2 to patients with idiopathic PAH to correct this imbalance was reported to result in a rapid decrease in pulmonary artery pressure and pulmonary vascular resistance and this finding led to the development of various therapeutic agents such as PGI2 derivatives [57]. In particular, epoprostenol is the only drug known to have a positive impact on long-term survival, and the prognosis of patients with PAH has improved dramatically since the clinical availability of continuous intravenous administration of epoprostenol [58].

Until now, the site of action of PGI2 agonists was initially thought to be only the IP receptor. However, long-term administration of PGI2 agonists downregulates IP receptor expression, but does not alter its efficacy in inhibiting pulmonary smooth muscle cell proliferation. Therefore, it has been suggested that the mechanism of action of PGI2 in pulmonary hypertensive smooth muscle cells is partially mediated by mechanisms other than IP receptor binding and cAMP production. In fact, PGI2 is less selective for IP receptors and can bind and activate other prostanoid receptors such as EP1, EP3, and TP receptors even with low affinity. Therefore, when IP receptors are decreased or their activity is inhibited, EP and TP receptor signals may be enhanced [59,60]. However, their activation tends to promote vascular remodeling in many cases, which cannot explain the antiproliferative effects of PGI2.

Another possible mechanism of action of PGI2 is the nuclear receptor proliferator-activated receptors (PPARs). PPARs are known to be activated by a variety of endogenous ligands, such as various fatty acids and their metabolites including prostaglandins. In addition, PPARs are thought to translocate into the nucleus by binding PGI2, function as transcription factors, and modulate cell growth, inflammation, and apoptosis. It has been noted that PGI2 analogs can directly bind to PPAR α or β and activate them as efficiently as endogenous and synthetic ligands [61]. In a study using pulmonary artery smooth muscle cells from patients with PAH, the antiproliferative effects of PGI2 analogs were independent of IP receptors and cAMP, but dependent on PPARγ [62].

#### 3.2.2. Leukotrien B4 (LTB4)

Alongside PGI2, LTB4 is one of the key lipid mediators responsible for promoting the pathogenesis of PH [63]. LTB4 is produced from AA as a substrate by LT synthase groups such as 5-LOX/5-LOX-activating protein (FLAP) and LTA4 hydrolase (LTA4H) and is closely related to several inflammatory diseases such as asthma and arteriosclerosis [64]. LTA4H is strongly expressed in perivascularly accumulated macrophages in lung tissue from PH rats (Sugen/athymic rat model) or patients with PAH [65]. Moreover, LTB4 inhibits the endothelial sphingosine kinase-1 (Sphk1)/eNOS pathway in pulmonary vascular endothelial cells to suppress apoptosis and induce abnormal proliferation and hypertrophy of vascular smooth muscle cells, thereby promoting pulmonary vascular remodeling. Inhibition of LTA4H or BLT1, the receptor for LTB4, has been shown to improve PH in animal models [65]. Furthermore, LTB4 also has the potential to directly affect lung fibroblasts, inducing proliferation, migration, and activation of the fibroblasts by enhancing p38-MAPK and NADPH oxidase 4 (Nox4) signaling pathway [66]. Based on these findings, clinical application targeting the LTB4 production system is expected. To verify the efficacy of the LTA4H inhibitor Bestatin (Ubenimex), a multi-center, double-blind Phase 2 trial (LIBERTY clinical trial, NCT02664558) was conducted in the United States in 2016, enrolling 61 patients with PAH, but unfortunately, its efficacy could not be proven. However, subsequent basic research revealed that induction of 5-LOX which catalyzes the formation of LTA4 that is hydrolyzed to LTB4 in bone morphogenic protein receptor-2 haploinsufficient (BMPR2+/−) rats resulted in severe PAH with pulmonary vascular remodeling. Moreover, that the expression of 5-LOX was increased in the neointimal cells of the pulmonary artery in BMPR2+/− rats or in patients with PAH with BMPR2 variants [67], indicating that LTB4 is an important mediator that drive the progression of PH as a factor of a ‘two hit’ model of PAH. In addition, RP5063, an antagonist for serotonin that has been linked to PH, reduces LTB4 in the PH rat model lung, suggesting crosstalk between serotonin and LTB4, and is expected as a new therapeutic target [68].

#### 3.2.3. Hydroxyeicosatetraenoic Acid (HETE)

5-hydroxyeicosatetraenoic acid (HETE), 12-HETE, and 15-HETE, which are produced from AA by LOX, are also known to be related to PH. These fatty acid metabolites are significantly increased in the lung or the serum from PH model animals or patients with severe PAH [69,70]. The administration of the 5-LOX inhibitor diethylcarbamazine improves PH in rat models [71]. 12-HETE is elevated in the lungs of hypoxia-exposed PH rats, enhancing the proliferation of smooth muscle cells via activation of ERK1/2 [72]. Additionally, 15-HETE has pathological activities such as hyperangiogenesis, smooth muscle cell proliferation, apoptosis resistance, and pro-fibrotic effects, suggesting a contribution to vascular remodeling [73,74,75,76].

#### 3.2.4. Epoxyeicosatrienoic Acid (EET)

Epoxyeicosatrienoic acid (EET) produced from AA by Cytochrome P450 epoxygenases (CYP2C, CYP2J) is also known to have multiple, strong bioactivities. Epoxides have a characteristic structure of three-membered ring ethers formed by the Cytochrome P450 system [77,78]. Intracellular levels of EETs are regulated by the activity of producing enzymes CYP epoxygenases and degrading enzyme soluble epoxide hydrolases that convert EETs to their corresponding diols. EETs are known to function as endothelium-derived vasodilators in the systemic vasculature, whereas in the pulmonary circulation, EETs are derived from vascular smooth muscle cells and promote vasoconstriction [79,80]. Intravenous administration of EETs increased right ventricular pressure and exacerbated PH [81]. In addition, EETs act on pulmonary endothelial cells to induce proliferation, apoptosis resistance, and angiogenesis through JNK/c-Jun activation, promoting pulmonary vascular remodeling [82]. Furthermore, 11,12-EET potentiates vasocontractile responses through transient receptor potential C6 channels [83]. Finally, pulmonary CYP epoxygenases and soluble epoxide hydrolase that regulate EETs level in lung play pivotal roles in the pulmonary vascular response to hypoxia in PH [84,85].

### 3.3. n-3 PUFAs and Their Derivatives

N-3 PUFAs, such as eicosapentaenoic acid (EPA) and docosahexaenoic acid (DHA), are known to exhibit beneficial effects on human health and their derivatives also acquire specific bioactivities to protectively modulate various diseases including cardiovascular diseases [86,87,88]. They regulate tissue remodeling to maintain homeostasis by suppressing abnormal activation of fibrosis and inflammation. For example, 18-HEPE, an EPA-derived metabolite, produced from macrophages in the heart, suppress abnormal activation of cardiac fibroblasts in a pressure overloaded heart, resulting in an improvement of cardiac remodeling [89]. In recent years, n-3 PUFA-derived metabolites, specialized pro-resolving mediators (SPMs), such as resolvin, protectin, and maresin, have been known to possess the potential to converge inflammation [90].

Indeed, several studies reported that n-3 PUFA or SPMs can improve PH in vivo and in vitro. The administration of EPA ameliorated right ventricular hypertrophy, remodeling and dysfunction, and medial wall thickening of the pulmonary arteries and prolonged survival in MCT-induced PAH rats [91]. Furthermore, in SPMs, resolvin D1 and resolvin E1 exhibit effects to reduce the contraction of human pulmonary arteries [92,93]. In addition, resolvin E1 was decreased in the serum of patients with idiopathic PAH and in the lung tissue of experimental rodent models of PH. Resolvin E1 acts on ChemR23 to inhibit the Wnt7a/β-Catenin pathway, thereby suppressing pulmonary artery smooth muscle cell proliferation and improving PH [94]. Resolvin E1 is also reported to improve vasoconstriction and vascular remodeling by inhibiting Src family kinases, especially Fyn, in human pulmonary artery endothelial cells and smooth muscle cells [91]. Maresin1, a representative SPM derived from DHA, also attenuated the phenotype of Sugen/hypoxia PH mice through inhibition of smooth muscle cell proliferation [95].

Thus, while the contribution of various lipid mediators in PH and pulmonary vascular remodeling is becoming clear, now there is little evidence of how mast cells, one of the sources of lipid mediators, are involved in PH. This should be considered an important research topic in the future.

## 4. Lipid Mediators Are Produced by Mast Cells Dependently or Independently of Degranulation

Various lipid mediators are released from mast cells activated by IgE. Representative lipid mediators, many of which are AA derivatives, include prostaglandin D2 (PGD2), leukotriene B4 (LTB4), LTC4, and platelet activating factor (PAF) [47,96]. Upon mast cell activation, phospholipase A2 (cPLA2α) moves from the cytoplasm to the perinuclear, Golgi. and endoplasmic reticulum membranes and cleaves AA from membrane phosphatidylcholine (PC) and phosphatidylethanolamine. Subsequently, LTC4 and LTB4 are produced by the 5-LOX pathway, and PGD2 is produced by the COX pathway. These are extracellularly secreted via active transporters and function as autocrine and paracrine mediators [96]. Of the mediators released from mast cells, PAF is an active phospholipid associated with allergy. Alkyl-lyso PC generated by cPLA2α is converted to PAF by the action of lysophosphatidylcholine acetyltransferase 2 (LPCAT2). The deletion of gene expression of cPLA2α, the producing enzymes and receptors for LT, PGD2, amd PAF in mice have exhibited a decline in asthmatic response, indicating the importance of mast cell-released lipid mediators in allergic diseases [47,97,98].

Recently, Shimanaka et al. found by using lipidomic analysis, that n-3 fatty acid epoxides (17,18-EpETE, 19,20-EpDPA) derived from EPA and DHA were more abundant than AA metabolites in the culture supernatant of unstimulated mast cells [99]. Moreover, they revealed that n-3 fatty acid epoxides were generated from epoxidized phospholipids contained in the plasma membrane phospholipids of mast cells through hydrolysis by type II platelet activating factor acetylhydrolase (PAF-AH2) and functioned intracellularly in the IgE-dependent degranulation [99]. Many n-3 fatty acid metabolites, such as SPMs, have been shown to have anti-inflammatory and bioprotective roles [88,100,101]. In contrast, n-3 fatty acid epoxides promote allergic inflammation mediated by mast cells. Although mast cell-derived lipid mediators have been generally thought to be secreted in response to external stimuli, it was recently revealed that epoxidized n-3 fatty acids are constitutively produced intracellularly and regulate the IgE/FcεRI-dependent activation of mast cells [47,99,102].

## 5. Novel Mast Cell-Derived Lipid Mediators in Pulmonary Hypertension: N-3 Fatty Acid Epoxides

As described above, evidence is accumulating that either mast cells or lipid mediators are involved in PH. There have been some reports on the effects of mast cell-derived lipid mediators on blood vessels (Table 1), but little is known about how they are involved in the pathogenesis of PH.

Recently, we revealed a novel pulmonary vascular remodeling regulatory mechanism in which mast cell-derived n-3 fatty acid epoxides contribute to PH pathogenesis without IgE stimulation or mast cell degranulation (Figure 2) [110]. Lipidomic analysis using LC-MS/MS for fatty acid metabolites found that n-3 fatty acid epoxides (17,18-EpETE, 19,20-EpDPA) and their diols (17,18-diHETE, 19,20-DiHDoPE) were decreased in PH lungs in response to hypoxia. In a hypoxia-exposed PH mouse model deficient in PAF-AH2, an enzyme that produces n-3 fatty acid epoxides, showed progressive pulmonary vascular remodeling, right heart failure, and worse survival rates compared to a hypoxic PH model in wild-type mice. In PH lungs from both mice and humans, pulmonary mast cells strongly expressing PAF-AH2 accumulated around pulmonary arteries. To selectively delete PAF-AH2 expression only in mast cells, we generated chimeric mice in which cultured bone marrow-derived mast cells were transplanted into mast cell-deficient Kit^W-sh/W-sh^ mice [111]. The chimeras showed exacerbation in hypoxia-induced PH to the same extent as in PAF-AH2 KO mice, demonstrating that mast cell-derived n-3 fatty acid epoxides modulate the severity of PH. Interestingly, the degranulation capacity of PAF-AH2-deficient mast cells was similar to that of normal mast cells under hypoxic stimulation. Treatment with n-3 fatty acid epoxides suppressed TGF-β-stimulated activation of lung fibroblasts, but did not affect vascular endothelial cells or smooth muscle cells, which are generally thought of as the main players in the pathogenesis of PH. Furthermore, in vivo administration of n-3 fatty acid epoxides inhibited the exacerbation of PH caused by PAF-AH2 deficiency and improved the PH phenotype even in Sugen/hypoxia mouse models more closely resembling the human pathology, demonstrating the potential of n-3 fatty acid epoxides as novel drug targets. Therefore, these results show a new regulatory mechanism of pulmonary vascular remodeling in PH, in which n-3 fatty acid epoxides produced by PAF-AH2 in mast cells suppress abnormal activation of fibroblasts. Furthermore, in hypoxic conditions, the expression of PAF-AH2 and the production of n-3 fatty acid epoxides in mast cells are reduced, resulting in overactivation of fibroblasts by TGF-β signaling, leading to the development of pulmonary vascular remodeling.

In addition, in this study, we examined the genomic information of patients with PAH for genetic variants in PAF-AH2 that are associated with the development of human PAH. Using whole-exome sequencing data from 262 patients with PAH, we identified two variants (R85C and Q184R) that were presumed to be highly pathogenic. When PAF-AH2 proteins with the same variants were overexpressed in vitro, the amount of variant PAF-AH2 proteins decreased, suggesting that the protein degradation was activated because of the change in protein conformation. In clinical practice, administration of n-3 fatty acid epoxides to PAH patients with these PAF-AH2 variants may be effective, and the development into precision medicine is anticipated.

## 6. Conclusions

Mast cells resident in tissues are strongly affected by the local environment, exerting various functions in each tissue in various pathological conditions and playing unique roles. In addition, the development of lipidomics technology has enabled accurate measurement of fatty acid metabolites, and there are reports on how bioactive lipids and their producing enzymes are involved in various pathological conditions [112]. For intractable diseases including PH, the fundamental pathological mechanisms have not yet been elucidated, and effective therapeutic materials are lacking. Bioactive lipids are greatly involved in pulmonary vascular remodeling, which is the pathological basis of PH, and many studies support the existence of candidate molecules among them as therapeutic targets. We identified n-3 fatty acid epoxides that can improve pulmonary vascular remodeling in PH and revealed their association with mast cells. In the future, with the accumulation of further evidence, treatments targeting mast cells or their specific production enzymes, direct administration of active lipids, and nutritional interventions aimed at supplementation with fatty acids may be novel therapeutic strategies for PH.

## Figures and Tables

**Figure 1 ijms-24-06619-f001:**
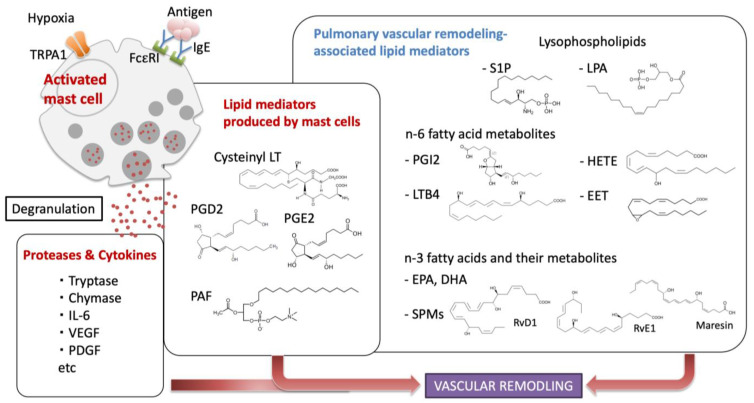
Mechanisms of vascular remodeling by mast cells and various lipid mediators. Abbreviations: TRPA1, transient receptor potential ankyrin 1; VEGF, vascular endothelial growth factor; PDGF, platelet-derived growth factor; LT, leukotriene; PGD2, prostaglandin D2; PGE2, prostaglandin E2; PAF, platelet activating factor; S1P, sphingosine-1 phosphate; LPA, lysophosphatidic acid; PGI2, prostaglandin I2; LTB4, leukotrien B4; HETE, hydroxyeicosatetraenoic acid; EET, epoxyeicosatrienoic acid; EPA, eicosapentaenoic acid; DHA, docosahexaenoic acid; SPMs, specialized pro-resolving mediators; RvD1, resolvin D1; RvE1, resolvin E1.

**Figure 2 ijms-24-06619-f002:**
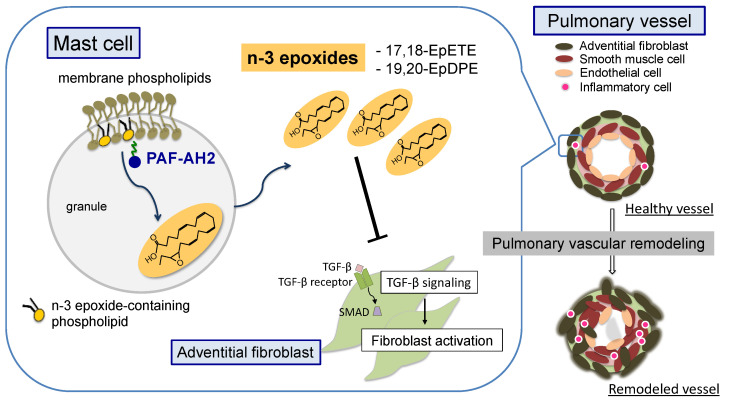
A novel mechanism of pulmonary vascular remodeling by n-3 epoxides. Mast cells-derived n-3 epoxides produced by PAF-AH2 regulate pulmonary vascular remodeling through inhibiting abnormal activation of adventitial fibroblasts. PAF-AH2, type II platelet activating factor acetylhydrolase.

**Table 1 ijms-24-06619-t001:** Representative mast cell-derived lipid mediators and the effects on vessels.

Mast Cell-Derived Lipid Mediator	Producing Enzyme	Effects on Vessels	References
PGD2	PGDS	Vasodilatation and promoting vascular permeability	[103,104]
PGD2	PGDS	Attenuating PH via CRTH2-mediated Th2 activation	[105]
PGE2	PGES	Promoting pulmonary vascular remodeling (in allergen-induced pulmonary inflammation)	[106]
Cysteinyl LT (LTC4, LTD4, LTE4)	LTC4S	Vasoconstriction under hypoxia	[107]
Cysteinyl LT (LTC4, LTD4, LTE4)	LTC4S	Promoting vascular permeability Sustained smooth muscle contraction	[47]
PAF	LPCAT2	Vasodilatation and promoting vascular permeability	[108,109]
n-3 epoxides (17,18-EpETE, 19,20-EpDPE)	PAF-AH2	Suppressing pulmonary vascular remodeling via inhibiting fibroblast activation	[110]

PGD2, prostaglandin D2; PGDS, prostaglandin D synthase; CRTH2, chemoattractant receptor-homologous molecule expressed on Th2 cells; PGE2, prostaglandin E2; PGES, prostaglandin E synthase; LT, leukotriene; LTC4S, leukotriene C4 synthase; PAF, platelet activating factor; LPCAT2, lysophosphatidylcholine acetyltransferase 2; PAF-AH2, type II platelet activating factor acetylhydrolase.

## Data Availability

Not applicable.

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
