# Peer review of "Pathophysiological Involvement of Mast Cells and the Lipid Mediators in Pulmonary Vascular Remodeling"

_ijms, 2023, doi:10.3390/ijms24076619_

Round 1

Reviewer 1 Report

In this manuscript (ID#: ijms-2220633), entitled “Pathophysiological Involvement of Mast Cells and the Lipid Mediators in Pulmonary Vascular Remodeling”. Authors, Moriyama et al, reviewed the role of Master cells and lipids metabolites in the development of pulmonary vascular remodeling, which is an important issue in the pulmonary hypertension. However, there are several major concerns, which are listed in the following paragraphs:

1. In page 2, line 50, “mast cells are less commonly found in the lung”, please clarify this is under physiological condition or pathophysiological condition. If mast cells are less in the lung, why they are important in the pulmonary vascular remodeling?

2. In page 2, lines 61-69, I do not know what this paragraph is talking about. Please verify if these are related to the topic of this review.

3. Please provide detail information regarding how mast cells are activated in pulmonary hypertension. Why the lipid metabolites produced within mast cells are elevated in pulmonary hypertension? It would be better to generate a summary table for all mast cell-derived lipid metabolites, producing enzymes, and their effects on pulmonary artery remodeling.

4 Fig 1 is confusing because it seems to me that lipid mediator is separated from mast cells. Please modify accordingly. In the blood vessel picture, please label all the cell types in the figure (I can see they are listed in Fig 2, maybe just copy and paste to Fig 1). Inflammatory cell is master cell (this review focusing on mast cells)?

5. Fig 2 needs to be modified. The adventitial fibroblasts in the figure should be outside of Mast cells

Reviewer 2 Report

This is an excellent review of the current knowledge related to the role of mast cells and bioactive lipids in the pathogenesis pulmonary hypertension. The work is a very nice addition to the field. Overall, the manuscript is well written. However, there are some minor issues to be addressed prior to acceptance of the manuscript for publication.

Lines 40-41. I would suggest to use “peripheral pulmonary artery lumen narrowing” instead of “stenosis”.

Lines 61-69. It seems that the authors included reviewer’s comments into the text.

Line 76. The phrase “underlying complications” makes no sense. Please improve it.

Line 80. “constituent cells” sounds awkward. Please improve it.

Line 116. PAH is a clinical group 1 according to the classification. So all animal models are experimental PH, not PAH.

Lines 351-355. There is a duplication of the text fragment in the same sentence, Please correct this.

Reviewer 3 Report

In the current manuscript Moriyama and Endo provide a comprehensive overview on the role of mast cells and lipid mediators in pulmonary vascular remodeling. Moreover , the illustrate how mast cells play a role in the regulation of these lipid mediators that influence PH. 

The manuscript is well structured and is quite complete in the view on mast cell and bioactive lipids function in pulmonary hypertension.

I only have a few minor points:

Line 61-69 need to be removed, these are instructions on how to prepare a manuscript and should have been removed.

In the introduction on pulmonary hypertension some words might be spend on the role of the TGF/BMP signaling pathway a one of the key drivers of PH. Just to illustrate it is not only mast cells and bioactive lipid mediators that regulate PH.

In the figures as there are in my version of the manuscript, some blocks appear with a black background (e.g. the background for the structure of S1P and HETE), which is not intended I suppose. Please check this for both figure 1 and figure 2.

Round 2

Reviewer 1 Report

The revised manuscript has been improved. No addition recommendation. 

Author Response

Thank you for your valuable comments.